# Novel Biologically Active *N*-Substituted Benzimidazole Derived Schiff Bases: Design, Synthesis, and Biological Evaluation

**DOI:** 10.3390/molecules28093720

**Published:** 2023-04-25

**Authors:** Anja Beč, Maja Cindrić, Leentje Persoons, Mihailo Banjanac, Vedrana Radovanović, Dirk Daelemans, Marijana Hranjec

**Affiliations:** 1Department of Organic Chemistry, Faculty of Chemical Engineering and Technology, University of Zagreb, Marulićev trg 19, HR-10000 Zagreb, Croatia; 2KU Leuven Department of Microbiology, Immunology and Transplantation, Rega Institute, 3000 Leuven, Belgium; 3Pharmacology In Vitro, Selvita Ltd., Prilaz baruna Filipovića 29, HR-10000 Zagreb, Croatia

**Keywords:** antiproliferative activity, antiviral activity, benzimidazoles, cytotoxicity, Schiff bases

## Abstract

Herein, we present the design and synthesis of novel *N*-substituted benzimidazole-derived Schiff bases, and the evaluation of their antiviral, antibacterial, and antiproliferative activity. The impact on the biological activity of substituents placed at the N atom of the benzimidazole nuclei and the type of substituents attached at the phenyl ring were examined. All of the synthesized Schiff bases were evaluated in vitro for their antiviral activity against different viruses, antibacterial activity against a panel of bacterial strains, and antiproliferative activity on several human cancer cell lines, thus enabling the study of the structure−activity relationships. Some mild antiviral effects were noted, although at higher concentrations in comparison with the included reference drugs. Additionally, some derivatives showed a moderate antibacterial activity, with precursor **23** being broadly active against most of the tested bacterial strains. Lastly, Schiff base **40,** a 4-*N*,*N*-diethylamino-2-hydroxy-substituted derivative bearing a phenyl ring at the N atom on the benzimidazole nuclei, displayed a strong antiproliferative activity against several cancer cell lines (IC_50_ 1.1–4.4 μM). The strongest antitumoral effect was observed towards acute myeloid leukemia (HL-60).

## 1. Introduction

Schiff bases comprise a very important class of organic compounds [1], widely used in organic and medicinal chemistry as biologically important structural motifs in many synthetic and semisynthetic organic compounds [2,3,4,5]. Some naturally occurring Schiff bases play important roles in several physiological processes, for example rhodopsin, a photoreceptor present in the rod cells of the retina, which is essential in vision processing [6]. Schiff bases can be easily synthesized by a condensation reaction of different amino substituted compounds with versatile aldehydes or ketones [7]. Besides the fact that Schiff bases are important building blocks in medicinal chemistry due to their broad spectrum of biological activity [8,9,10,11,12], they have also been used in coordination chemistry [13,14,15], in analytical chemistry [16], as dyes [17], as optical chemosensors [18,19,20], as polymers [21], in catalysis [22], in metallurgy and refining of metals, and as fungicidal and agrochemical compounds. Additionally, the interest of numerous scientists is focused on evaluating Schiff bases as ligands for transition metals displaying diverse biological activities, such as anticancer [23,24], antimicrobial [25,26], and antifungal activities [27].

There is growing interest in the synthesis of benzimidazole-derived Schiff bases, not only because of their significant biological activities, but also because they can easily form complexes with different metals, giving rise to a variety of complexes with interesting electronic [28,29] and biological properties [30].

Nawrocka et al. synthesized novel 2-benzimidazolyl-substituted Schiff bases that were screened for their antiproliferative activity on several cancer cell lines [31]. Another group designed and synthesized benzimidazole and 3-oxopyrimido [1,2-*a*]benzimidazole-derived Schiff bases being evaluated as inhibitors of lipoxygenase (LOX) and of lipid peroxidation (LPO), showing antioxidative and cytotoxic activity (Figure 1a) [32]. In another study, a series of Schiff bases bearing benzimidazole nuclei were evaluated for their antimalarial and antitrypanosomal activity [33]. Furthermore, a group of authors explored the antimicrobial activity of a series of benzimidazole-derived Schiff bases against *Staphylococcus aureus* and *Escherichia coli* (Figure 1b) [34]. The metal–Schiff-base complexes could bind ct-DNA through intercalation and were found to exhibit more potent cytotoxic effects than the widely used drug cisplatin [35]. Another group of authors developed metal complexes that were evaluated for their cytotoxic, antiparasitic, and antibacterial activity, as well as their interaction with DNA [36]. Other authors studied the DNA interaction and antiproliferative activity of two Cu(II) complexes with Schiff bases of benzimidazole [37].

Recently, we synthesized a series of novel benzimidazole-substituted Schiff bases (Figure 2a) that were tested in vitro for their antiproliferative activity. They exerted a broad spectrum antiproliferative activity on different cancer cell lines, although only at higher concentrations [38]. Compounds substituted with 4-*N*,*N*-diethylamino-2-hydroxyphenyl bearing either a methyl or a cyano group at the 5(6)-position on the benzimidazole nuclei displayed the strongest antiproliferative effect on all of the tested cell lines, with a significant concentration-dependent effect on HeLa and MCF-7 cell lines.

As a continuation, we designed and synthesized novel *N*-substituted benzimidazole-derived Schiff bases (Figure 2b) to explore the influence of the substituents attached at the N atom on the benzimidazole nuclei, both for their antiproliferative and antiviral activity. Our main goal was to see how the different lengths and types of aliphatic chain and the aromatic moiety attached to the nitrogen atom on the benzimidazole nuclei would influence the biological activity.

## 2. Results and Discussion

### 2.1. Chemistry

The synthesis of the novel *N*-substituted benzimidazole-derived Schiff bases **28**–**45** is presented in reaction Figure 1. For the synthesis of the targeted Schiff bases, as main precursors, corresponding *N*-substituted 2-aminobenzimidazoles **17**–**24** were prepared according to the modified reaction procedures previously described by our research group. Starting from 2-chloronitrobenzene **1** or 2-chloro-4-cyanonitrobenzene **2**, in the reaction of uncatalyzed microwave-assisted amination with an excess of amine, *N*-methyl **5**, *N*-phenyl **6**, *N*-isobutyl **3**–**4,** and *N*-hexyl **7–8** substituted nitro precursors obtained in moderate yields. After reduction with SnCl_2_ × 2H_2_O in acidic media, corresponding 1,2-diamino substituted benzenes **9**–**16** were obtained. Because of the cyclocondensation with cyanogen bromide in methanol, substituted 2-aminobenzimidazoles with isobuthyl **17** and **21**, methyl **18** and **22**, phenyl **19** and **23,** or *n*-hexyl **20** and **24** side chains at the N atom or cyano group placed at the 5(6)-position on the benzimidazole nuclei were prepared in moderate reaction yields.

Benzimidazole-derived Schiff bases **28**–**45** were synthesized from *N*-substituted 2-aminobenzimidazoles **17**–**24** with chosen substituted benzaldehydes, namely 4-*N*,*N*-dimethylamino-**25**, 2-hydroxy-4-*N*,*N*-dimethylamino-**26,** and 4-nitrobenzaldehyde **27,** according to the previously published experimental procedure [38]. All of the Schiff bases were additionally purified either by recrystallization or by using column chromatography on silica gel with dichloromethane/methanol as an eluent to yield the targeted compounds in low to moderate reaction yields (6–68%).

The structures of all newly prepared Schiff bases **28**–**45** were confirmed by means of ^1^H and ^13^C NMR spectroscopy and elemental analysis. Structural analysis was performed based on the chemical shifts in both the proton spectra and ^13^C NMR spectra, and on the values of H–H coupling constants in the proton spectra. The amination of reactants **1–2** caused the appearance of signals related to the amino group (8.12–9.90 ppm), as well as the signals for protons from isobutyl, methyl, and *n*-hexyl sidechains (0.87–3.41 ppm) in the structure of compounds **3–8**. Reduction of the nitro group was confirmed within the singlet related to the protons from the amino group (4.44 to 5.25 ppm). The formation of the benzimidazole ring was confirmed by the disappearance of signals for the amino groups in 1,2-diamino-substituted benzenes **9–16,** as well as the appearance of a singlet related to the proton of the amino group placed at position 2 on the benzimidazole nuclei (**17**–**24**), which was downshifted in comparison with the singlet of the amino group in the diamino-substituted benzenes **9**–**16**. The synthesis of targeted Schiff bases **28**–**45** was established by the observation of a singlet related to the proton of the imino group at 7.98–9.67 ppm in the proton spectra, as well as a signal for the C atom of the imino group in the carbon spectra.

### 2.2. Biological Activity

#### 2.2.1. Cytotoxicity and Antiviral Activity

In Table 1, the summarized results of the antiviral evaluation are depicted. For a better overview, only the derivatives showing antiviral activity against this panel of viruses are included.

The following viruses were used for evaluating the antiviral activity: different strains of human coronavirus, three influenza virus subtypes, RSV, HSV-1, yellow fever virus, Zika virus, and Sindbis virus. The results are expressed as CC_50_ (50% cytotoxic concentration) and EC_50_ (50% effective concentration) values. Overall, the *N*-substituted 2-aminobenzimidazoles **17**–**24** displayed a poor antiviral activity. Substituted 2-aminobenzimidazoles bearing methyl **18** and **22** were both able to inhibit Zika virus replication in Huh-7 cells with EC_50_ values of 43.1 µM and 46.4 µM, respectively. Among the *N*-substituted benzimidazole-derived Schiff bases **28**–**45,** some derivatives showed an antiviral activity, although it was weak when compared with the standard antiviral drugs included (remdesivir, ribavirin, zanamivir, rimantadine, and BVDU). The 4-*N*,*N*-diethylamino-2-hydroxy-substituted derivative bearing a cyano and *N*-isobutyl side chain on the benzimidazole nuclei **31** showed a moderate activity against HCoV-NL63 in Huh-7 cells (EC_50_ 32 μM). Compound **42**, substituted with a *N*,*N*-dimethylamino and a *N*-hexyl side chain showed a moderate activity against HCoV-229E in HEL299 cells (EC_50_ 34.7 μM). In summary, no outstanding antiviral effects were noted for the 4-*N*,*N*-diethylamino-2-hydroxyphenyl ring nor for the 4-*N*,*N*-dimethylaminophenyl and 4-nitrophenyl rings.

#### 2.2.2. Antibacterial Activity

The in vitro antibacterial activity of the synthesized Schiff bases was evaluated against a panel of eight different bacterial strains. Gram-positive bacterial strains comprised *S. aureus*, S*. pneumoniae* and *E. faecalis* and the panel of Gram-negative bacteria consisted of *E. coli*, *K. pneumoniae*, *A. baumannii,* and *P. aeruginosa*. As reference drugs, the antibiotics ampicillin, ceftazidime, ciprofloxacin, and meropenem were included.

As presented in Table 2, the majority of derivatives lacked an antibacterial activity, while some Schiff bases showed a moderate activity against certain bacterial strains.

The *N*,*N*-dimethylamino **38** and 4-*N*,*N*-diethylamnino-2-hydroxy **41** derivatives, both substituted with a phenyl ring at the N atom on the benzimidazole nuclei, displayed activity against *E. faecalis* (MIC 32 μM). *p*-Nitro and cyano-substituted Schiff base **37** with a methyl group at the N atom showed activity against *S. aureus* (32 μM). *N*,*N*-dimethylamino derivative **42** with a hexyl side chain placed at the N atom demonstrated a moderate activity against *S. aureus* and *E. faecalis* (32 μM). Overall, precursor *N*-hexyl-2-aminobenzimidazole **23** showed the most pronounced broad spectrum of antibiotic activity against *E. faecalis*, *E. coli efflux dell*, and *K. pneumoniae,* with MIC values of 32 μM, and against *S. aureus* with an MIC value of 16 μM.

#### 2.2.3. Antiproliferative Activity

All of the prepared *N*-substituted 2-aminobenzimidazoles **9**–**16** and benzimidazole-derived Schiff bases **28**–**45** were explored for their in vitro antiproliferative activity against several human cancer cell lines. The results are presented in Table 3 as IC_50_ values (50% inhibitory concentration). The following human cancer cell lines were used for the evaluation of the antiproliferative activity: LN-229, glioblastoma; Capan-1, pancreatic adenocarcinoma; HCT-116, colorectal carcinoma; NCI-H460, lung carcinoma; DND-41, acute lymphoblastic leukemia; HL-60, acute myeloid leukemia; K-562, chronic myeloid leukemia; and Z-138, non-Hodgkin lymphoma cancer cells. All of the obtained results were compared to *etoposide* (ETP) and *nocodazole* (NZO) as the standard chemotherapeutic agents.

The majority of tested compounds showed low or no activity towards the selection of cancer cell lines (for clarity, some inactive 2-aminobenzimidazoles are excluded from Table 3). Among all of the tested *N*-substituted 2-aminobenzimidazoles, the best activity was demonstrated by *n*-hexyl-substituted derivative **23,** which showed a moderate but broad antiproliferative activity against all of the tested cancer types.

*N*-phenyl-substituted 2-aminobenzimidazole **21** showed a mild but selective activity against lung carcinoma. Regarding the benzimidazole-derived Schiff bases, the most potent one was the 4-*N*,*N*-diethylamino-2-hydroxy-substituted Schiff base bearing a phenyl ring at the N atom on benzimidazole nuclei **40**. This derivative displayed a pronounced antitumoral activity against Capan-1, DND-41, HL-60, and Z-138 cancer cells in the low micromolar range, with some selectivity towards the HL-60 (acute myeloid leukemia) cancer cell line (IC_50_ 1.1 µM). In addition, derivative **40** showed moderate activity against all of the other cancer cells lines. Furthermore, the 4-*N*,*N*-dimethylamino-substituted Schiff base bearing an isobutyl side chain at the N atom **28** showed a selective antiproliferative activity against colorectal carcinoma (HCT-116). Derivative **31** substituted with 2-hydroxy and 4-*N*,*N*-diethylamino groups at the phenyl ring, as well as with an isobutyl sidechain on the N atom and a cyano group on the benzimidazole nuclei, displayed a moderate activity against three cancer cell types (IC_50_ = 32.8–47.6 mM). The 4-*N*,*N*-diethylamino-2-hydroxy-substituted Schiff base **41** bearing a phenyl ring at the N atom and a cyano group on the benzimidazole nuclei also proved to be moderately active against several of the tested cancer cell lines. Among the *N*-hexyl-substituted Schiff bases **42**–**45**, a moderate activity was observed for compound **42** substituted with a 4-*N*,*N*-dimethylamino group at the phenyl ring, as well as for compound **43,** which also bears a cyano group at the benzimidazole nuclei.

The 4-*N*,*N*-diethylamino-2-hydroxy-substituted derivatives **44**–**45** were less active in comparison with the 4-*N*,*N*-dimethylamino substituted analogues. Regarding the *N*-methyl-substituted Schiff bases, only the 4-*N*,*N*-diethylamino-2-hydroxy-substituted derivative bearing a cyano group showed a moderate but selective activity against DND-41 cells. When comparing all of the *N*-isobutyl substituted Schiff bases **28**–**31**, we can conclude that the derivative bearing both a 4-*N*,*N*-diethylamino-2-hydroxyphenyl and cyano group **31** was the most active one.

To conclude, we observed that the most significant impact on the antiproliferative activity was seen for the 4-*N*,*N*-diethylamino-2-hydroxyphenyl ring in comparison with the 4-*N*,*N*-dimethylaminophenyl and 4-nitrophenyl rings. A cyano group placed at the 5(6) position on the benzimidazole nuclei increased the antiproliferative activity, but not for the *N*-phenyl-substituted derivatives. The 4-*N*,*N*-diethylamino-2-hydroxy-substituted Schiff base bearing a phenyl ring at the N atom on benzimidazole nuclei **40**, which showed the most promising activity among all of the tested derivatives, was chosen as a lead compound for further optimization in order to obtain more selective and potent antiproliferative agents (Figure 3).

## 3. Conclusions

Herein, we present the design and synthesis of novel, *N*-substituted benzimidazole-derived Schiff bases, bearing isobutyl, methyl, and *n*-hexyl sidechains or a phenyl group at the N atom on the benzimidazole nuclei as well as a 4-*N*,*N*-dimethylamino- or 4-*N*,*N*-diethylamino-2-hydroxyphenyl ring attached directly to the imino bond. Within this research, our main focus was to study the impact on the biological activity of the substituents placed on the phenyl ring and on the N atom of the benzimidazole nuclei. All Schiff bases were evaluated for their in vitro antiviral activity against a broad selection of viruses, antibiotic activity against Gram-positive and Gram-negative bacterial strains, and antiproliferative activity on a diverse panel of human cancer cell lines.

The series of newly synthesized derivatives lacked a pronounced antiviral activity against the panel of selected viruses. Schiff base **31** substituted with a 4-*N*,*N*-diethylamino-2-hydroxyphenyl and bearing a cyano and *N*-isobutyl side chain on the benzimidazole nuclei showed a moderate activity against HCoV-NL63 virus on Huh-7 cells, although with a low selectivity (EC_50_ 32 μM and CC_50_ 85.7 µM).

The majority of the tested compounds were inactive against the Gram-positive and Gram-negative bacterial strains used. A selection of Schiff bases showed a moderate activity against one and/or two bacterial strains, while the most active compound was *N*-hexyl-2-aminobenzimidazole **23** with a moderate but broad activity against *E. faecalis*, *E. coli* (*efflux dell*), and *K. pneumoniae* (MIC 32 μM), as well as against *S. aureus* (MIC 16 μM).

Additionally, all of the prepared 2-aminobenzimidazoles **17**–**24** and Schiff bases **28**–**45** were tested for their antiproliferative activity against several cancer cell lines. The obtained results revealed that Schiff base **31** substituted with a 4-*N*,*N*-diethylamino -2-hydroxyphenyl and with an isobutyl side chain on the N atom and a cyano group on the benzimidazole nuclei displayed a moderate activity against three cancer types (IC_50_ 32.8–47.6 µM). In addition, among the tested *N*-substituted-2-aminobenzimidazoles, *N*-hexyl-substituted derivative **23** showed a broad but modest activity against all of the tested cancer cell lines. Furthermore, the 4-*N*,*N*-diethylamino-2-hydroxyphenyl ring as well as the phenyl ring attached to the N atom on the benzimidazole nuclei had a pronounced impact on the activity. Schiff base **40** bearing the above-mentioned substituents showed the most promising selective antiproliferative activity against Capan-1, DND-41, HL-60, and Z-138 cancer cell lines (IC_50_ = 1.1–4.4 μM), and a moderate activity against all of the other tested cell lines.

In conclusion, we have shown that out of the prepared *N*-substituted benzimidazole-derived Schiff bases, derivative **40** showed the most interesting biological potential, with a promising antiproliferative activity, making it a promising candidate for further design and optimization.

## 4. Materials and Methods

### 4.1. General Methods

All of the chemicals were purchased from commercial suppliers. Melting points were recorded on Büchi 535 melting apparatus (Büchi, Sankt Gallen, Switzerland). The ^1^H and ^13^C NMR spectra were recorded on a Varian Bruker Advance III HD 400 MHz/54 mm Ascend instrument (Bruker, Billerica, MA, USA). All of the NMR spectra were measured in DMSO-d6 solutions using TMS as an internal standard. All of the compounds were routinely checked by TLC with Merck silica gel 60F-254 glass plates and the spots were detected under UV light. Microwave-assisted synthesis was performed in a Milestone start S microwave (Milestone Srl, Sorisole, Italy) oven using quartz cuvettes under a pressure of 40 bar. Elemental analyses for carbon, hydrogen, and nitrogen were performed on a PerkineElmer 2400 elemental analyzer (Perkin-Elmer, Waltham, MA, USA, SAD). Where analyses are indicated only as symbols of elements, the analytical results obtained were within 0.4% of the theoretical value. NMR spectra of synthesized compounds are given in Appendix A. 

### 4.2. Synthesis

#### 4.2.1. General Method for Preparation of Compounds **7**–**8**

Compounds **7**–**8** were prepared using microwave irradiation, at an optimized reaction time at 170 °C with power 800 W and 40 bar pressure, from 1 or 2 in acetonitrile (10 mL) with an excess of added corresponding amine. After cooling, the resulting product was purified by column chromatography on SiO_2_ using dichlormethane/methanol as the eluent.

##### *N*-Hexyl-2-nitroaniline **7**

**7** was prepared from **1** (0.50 g, 3.2 mmol) and hexylamine (2.90 mL, 22.2 mmol) after 2 h of irradiation to yield 0.60 g (94%) of orange oil. ^1^H NMR (400 MHz, DMSO-d_6_) (δ/ppm): 8.12 (t, 1H, J = 5.07 Hz, NH), 8.06 (dd, 1H, J_1_ = 1.58 Hz, J_2_ = 8.57 Hz, H_arom_), 7.56–7.51 (m, 1H, H_arom_), 7.05 (d, 1H, J = 7.99 Hz, H_arom_), 6.70–7.65 (m, 1H, H_arom_), 3.37–3.33 (m, 2H, CH_2_), 1.67–1.58 (m, 2H, CH_2_), 1.41–1.27 (m, 6H, CH_2_), 0.87 (t, 3H, J = 7.06 Hz, CH_3_); ^13^C NMR (151 MHz, DMSO-d_6_): δ/ppm = 145.7, 137.1, 131.3, 126.7, 115.5, 115.0, 42.7, 31.4, 28.7, 26.5, 22.5, 14.4; Anal. Calcd. for C_12_H_18_N_2_O_2_: C, 64.84; H, 8.16; N, 12.60; O, 14.39. Found: C, 64.89; H, 8.23; N, 12.51; O, 14.29%.

##### 3-*N*-(Hexylamino)-4-nitrobenzonitrile **8**

**8** was prepared from **2** (0.50 g, 2.7 mmol) and hexylamine (1.80 mL, 13.7 mmol) after 2 h of irradiation to yield 0.50 g (97%) of yellow oil. ^1^H NMR (400 MHz, DMSO-d_6_) (δ/ppm): 8.58 (t, 1H, J = 5.38 Hz, NH), 8.50 (d, 1H, J = 2.00 Hz, H_arom_), 7.81 (dd, 1H, J_1_ = 1.60 Hz, J_2_ = 9.08 Hz, H_arom_), 7.18 (d, 1H, J = 9.10 Hz, H_arom_), 3.41 (q, 2H, J = 6.72 Hz, CH_2_), 1.65–1.55 (m, 2H, CH_2_), 1.38–1.23 (m, 6H, CH_2_), 0.87 (t, 3H J = 6.70 Hz, CH_3_); ^13^C NMR (75 MHz, DMSO-d_6_): δ/ppm = 146.8, 130.5, 118.2, 96.0, 42.3, 30.8, 27.9, 25.8, 21.9; Anal. Calcd. for C_13_H_17_N_3_O_2_: C, 63.14; H, 6.93; N, 16.99; O, 12.94. Found: C, 63.20; H, 6.88; N, 16.91; O, 12.89%.

#### 4.2.2. General Method for Preparation of Compounds **15**–**16**

Derivative **7** or benzonitrile derivative **8** and a solution of SnCl_2_ × 2H_2_O in MeOH and concentrated HCl were refluxed for 0.5 h. The resulting solution was treated with 20% NaOH to pH = 14. The resulting precipitate was filtered off, washed with hot ethanol, and filtered. The filtrate was evaporated at a reduced pressure and was extracted with ethyl acetate. The organic layer was dried over anhydrous MgSO_4_ and concentrated at a reduced pressure.

##### *N*^1^-Hexylbenzene-1,2-diamine **15**

Compound **15** was prepared from **7** (3.52 g, 15.8 mmol), SnCl_2_ × 2H_2_O (29.70 g, 131.5 mmol), HCl_conc._ (49 mL), and MeOH (49 mL) to yield 2.13 g (70%) of red oil. ^1^H NMR (600 MHz, DMSO-d_6_) (δ/ppm): 6.52 (dd, 1H, J_1_ = 1.46 Hz, J_2_ = 7.78 Hz, H_arom_), 6.48 (td, 1H, J_1_ = 1.48 Hz, J_2_ = 7.56 Hz, H_arom_), 6.40–6.37 (m 2H, H_arom_), 4.44 (bs, 2H, NH_2_), 4.28 (t, 1H, J = 5.18 Hz, NH), 2.99 (q, 2H, J = 6.88 Hz, CH_2_), 1.61–1.55 (m, 2H CH_2_), 1.41–1.35 (m, 2H, CH_2_), 1.32–1.28 (m, 4H, CH_2_), 0.88 (t, 3H, J = 6.98 Hz, CH_3_); ^13^C NMR (151 MHz, DMSO-d_6_): δ/ppm = 136.1, 135.0, 117.5, 116.5, 114.0, 109.6, 43.4, 31.2, 28.8, 26.5, 22.1, 13.9; Anal. Calcd. for C_12_H_20_N_2_: C, 74.95; H, 10.48; N, 14.57. Found: C, 74.88; H, 10.54; N, 14.66%.

##### 3-Amino-4-*N*-(hexylamino)benzonitrile **16**

Compound **16** was prepared from **8** (2.71 g, 10.9 mmol), SnCl_2_ × 2H_2_O (14.85 g, 68.8 mmol), HCl_conc._ (29 mL), and MeOH (29 mL) to yield 1.65 g (69%) of pink powder. m.p. 157–161 °C; ^1^H NMR (600 MHz, DMSO-d_6_) (δ/ppm): 6.91 (dd, 1H, J_1_ = 1.90 Hz, J_2_ = 8.18 Hz, H_arom_), 6.76 (d, 1H, J = 2.00 Hz, H_arom_), 6.44 (d, 1H, J = 8.16 Hz, H_arom_), 5.32 (t, 1H, J = 5.00 Hz, NH), 4.96 (s, 2H, NH_2_), 3.08 (q, 2H, J = 6.42 Hz, CH_2_), 1.64–1.51 (m, 2H, CH_2_), 1.41–1.25 (m, 6H, CH_2_), 0.88 (t, 3H, J = 6.58 Hz, CH_3_); ^13^C NMR (75 MHz, DMSO-d_6_): δ/ppm = 140.5, 135.5, 123.4, 121.6, 115.2, 108.8, 96.7, 43.2, 31.6, 28.8, 26.8, 22.6, 14.4; Anal. Calcd. for C_13_H_19_N_3_: C, 71.85; H, 8.81; N, 19.34. Found: C, 71.94; H, 8.87; N, 19.41%.

#### 4.2.3. General Method for Preparation of Compounds **23** and **24**

BrCN was added dropwise to a solution of *o*-phenylenediamine or in 20 mL H_2_O and 5 mL acetonitrile. The reaction mixture was refluxed for 2 h and NH_4_OH was added to adjust to pH = 9. After cooling, the resulting precipitate was filtered off.

##### 2-Amino-1-hexylbenzimidazole **23**

Compound **23** was prepared from **15** (1.59 g, 8.3 mmol) and BrCN (0.88 g, 8.3 mmol) to to yield 1.45 g (81%) of brown powder. m.p. 245–250 °C; ^1^H NMR (600 MHz, DMSO-d_6_) (δ/ppm): 7.96 (bs, 2H, NH2), 7.43–7.41 (m, 1H, Harom), 7.32–7.30 (m, 1H, Harom), 7.18–7.14 (m, 2H, Harom), 4.08 (t, 2H, J = 7.37 Hz, CH2), 1.68–1.63 (m, 2H, CH2), 1.33–1.24 (m, 6H, CH2), 0.84 (t, 3H, J = 7.07 Hz, CH3); ^13^C NMR (151 MHz, DMSO-d_6_): δ/ppm = 151.6, 131.9, 122.9, 121.9, 112.9, 109.9, 42.4, 31.3, 28.2, 26.0, 22.5, 14.3; Anal. Calcd. for C_13_H_19_N_3_: C, 71.85; H, 8.81; N, 19.34. Found: C, 71.87; H, 8.79; N, 19.29%.

##### 2-Amino-6-cyano-1-hexylbenzimidazole **24**

Compound **24** was prepared from **16** (0.03 g, 1.4 mmol) and BrCN (0.14 g, 1.4 mmol) to to yield 0.28 g (83%) of grey powder. m.p. 215–220 °C; ^1^H NMR (600 MHz, DMSO-d_6_) (δ/ppm): 7.49 (s, 1H, H_arom_), 7.33–7,28 (m, 2H, H_arom_), 6.84 (s, 1H, H_arom_), 4.01 (t, 2H, J = 7.18 Hz, CH_2_), 1.63–1.58 (m, 2H, CH_2_), 1.27–1.22 (m, 6H, CH_2_), 0.83 (t, 3H, J = 6.68 Hz, CH_3_); ^13^C NMR (151 MHz, DMSO-d_6_): δ/ppm = 157.1, 143.2, 138.3, 121.2, 102.5, 42.1, 31.4, 28.7, 26.1, 22.5; Anal. Calcd. for C_14_H_18_N_4_: C, 69.39; H, 7.49; N, 23.12. Found: C, 69.45; H, 7.40; N, 23.07%.

#### 4.2.4. General Method for Preparation of Schiff Bases **28**–**45**

Solutions of the equimolar amounts of the corresponding *N*-substituted-2-aminobenzimidazole and aromatic aldehyde in absolute ethanol were refluxed for 24–48 h. After cooling, the obtained products were filtered off and recrystallized from ethanol. If necessary, the products were purified by column chromatography on SiO_2_ using a gradient elution of dichloromethane/methanol/TEA. Basic TEA was used to prevent the decomposition of the Schiff base conjugates in the silica gel column.

##### (*E*)-4-(((1-Isobutyl-1*H*-benzo[*d*]imidazol-2-yl)imino)methyl)-*N*,*N*-dimethylaniline **28**

Compound **28** was prepared from 2-amino-1-isobutylbenzimidazole **17** (0.10 g, 0.5 mmol) and 4-*N*,*N*-dimethylamino-benzaldehyde **25** (0.08 g, 0.5 mmol) in absolute ethanol (3 mL) after refluxing for 48 h to obtain 0.02 g (14%) of yellow powder. m.p. 277–279 °C; ^1^H NMR (600 MHz, DMSO-d_6_) (δ/ppm): 9.26 (s, 1H, H_arom_), 7.91 (d, 2H, J = 8.79 Hz, H_arom_), 7.55–7.50 (m, 2H, H_arom_), 7.19–7.14 (m, 2H, H_arom_), 6.85 (d, 2H, J = 8.88 Hz, H_arom_), 4.17 (d, 2H, J = 7.27 Hz, CH_2_), 3.07 (s, 6H, CH_3_), 2.24–2.16 (m, 1H, CH), 0.89 (t, 6H, J = 6.66 Hz, CH_3_); ^13^C NMR (151 MHz, DMSO-d_6_): δ/ppm = 150.4, 131.2 (2C), 129.1 (2C), 124.0, 123.4, 112.0 (2C), 111.2 (3C), 49.1, 27.7, 19.7 (2C); Anal. Calcd. for C_20_H_24_N_4_: C, 74.97; H, 7.55; N, 17.48. Found: C, 74.95; H, 7.59; N, 17.41%.

##### (*E*)-2-((4-(Dimethylamino)benzylidene)amino)-1-isobutyl-1*H*-benzo[*d*]imidazole-6-carbonitrile **29**

Compound **29** was prepared from 2-amino-6-cyano-1-isobutylbenzimidazole **21** (0.10 g, 0.5 mmol) and 4-*N*,*N*-dimethylamino-benzaldehyde **25** (0.07 g, 0.5 mmol) in absolute ethanol (4 mL) after refluxing for 24 h to obtain 0.03 g (22%) of yellow powder in the form of a mixture of *E*- and *Z*-isomers at a ratio of **29a**/**29b** = 5:3. m.p. 283–285 °C; **29a**: ^1^H NMR (400 MHz, DMSO-d_6_) (δ/ppm): 9.28 (s, 1H, H_arom_), 8.03 (d, 1H, J = 1.01 Hz, H_arom_), 7.94 (d, 2H, J = 8.78 Hz, H_arom_), 7.74 (d, 1H, J = 8.37 Hz, H_arom_), 7.69 (d, 1H, J = 9.02 Hz, H_arom_), 7.57 (dd, 1H, J_1_ = 8.31, J_2_ = 1.38 Hz, H_arom_), 6.85 (d, 2H, J = 9.03 Hz, H_arom_), 4.21 (d, 2H, J = 7.27 Hz, CH_2_), 3.08 (s, 6H, CH_3_), 2.23–2.16 (m, 1H, CH), 0.89 (d, 6H, J = 6.66 Hz, CH_3_); ^13^C NMR (100 MHz, DMSO-d_6_) (δ/ppm): 166.2, 159.0, 154.3, 141.6, 138.7, 132.6, 125.2, 122.7, 120.6, 112.1, 104.1, 49.8, 29.3, 20.3; **29b**: ^1^H NMR (400 MHz, DMSO-d_6_) (δ/ppm): 9.67 (s, 1H, H_arom_), 7.69 (d, 1H, J = 9.01 Hz, H_arom_), 7.49 (d, 1H, J = 1.09 Hz, H_arom_), 7.33 (d, 1H, J = 8.07 Hz, H_arom_), 7.28 (dd, 1H, J_1_ = 8.11, J_2_ = 1.37 Hz, H_arom_), 6.84–6.78 (m, 3H, H_arom_), 3.85 (d, 2H, J = 7.56 Hz, CH_2_), 3.05 (s, 6H, CH_3_), 2.13–2.04 (m, 1H, CH), 0.86 (d, 6H, J = 6.67 Hz, CH_3_); ^13^C NMR (100 MHz, DMSO-d_6_) (δ/ppm): 190.3, 157.4, 143.2, 132.0, 121.2, 111.9, 111.5, 109.3, 102.4, 48.9, 28.4, 20.0; Anal. Calcd. for C_21_H_23_N_5_: C, 73.02; H, 6.71; N, 20.27. Found: C, 73.10; H, 6.68; N, 20.32%.

##### (*E*)-5-(Diethylamino)-2-(((1-isobutyl-1*H*-benzo[*d*]imidazol-2-yl)imino)methyl)phenol **30**

Compound **30** was prepared from 2-amino-1-isobutylbenzimidazole **17** (0.15 g, 0.8 mmol) and 4-*N*,*N*-diethylamino-2-hydroxybenzaldehyde **26** (0.12 g, 0.8 mmol) in absolute ethanol (6 mL) after refluxing for 48 h to obtain 0.10 g (36%) of yellow powder. m.p. 254–258 °C; ^1^H NMR (300 MHz, DMSO-d_6_) (δ/ppm): 12.79 (s, 1H, OH), 9.35 (s, 1H, H_arom_), 7.55–7.50 (m, 2H, H_arom_), 7.42 (d, 1H, J = 8.97 Hz, H_arom_), 7.20–7.15 (m, 2H, H_arom_), 6.42 (dd, 1H, J_1_ = 2.27 Hz, J_2_ = 8.87 Hz, H_arom_), 6.17 (d, 1H, J = 2.09 Hz, H_arom_), 4.06 (d, 2H, J = 7.18 Hz, CH_2_), 3.46 (q, 4H, J = 7.08, CH_2_), 2.22–2.12 (m, 1H, CH), 1.12 (t, 6H, J = 7.09, CH_3_), 0.91 (d, 6H, J = 6.55 Hz, CH_3_); ^13^C NMR (151 MHz, DMSO-d_6_): δ/ppm = 163.5, 153.8, 152.9, 135.0, 121.8, 121.3, 117.9, 111.2, 110.0, 108.5, 105.0, 104.4, 96.6, 95.9, 49.4, 44.1 (2C), 28.9, 19.9 (2C), 12.5 (2C); Anal. Calcd. for C_22_H_28_N_4_O: C, 72.50; H, 7.74; N, 15.37; O, 4.39. Found: C, 72.48; H, 7.70; N, 15.32; O, 4.44%.

##### (*E*)-2-((4-(Diethylamino)-2-hydroxybenzylidene)amino)-1-isobutyl-1*H*-benzo[*d*]imidazole-6-carbonitrile **31**

Compound **31** was prepared from 2-amino-6-cyano-1-isobutylbenzimidazole **21** (0.15 g, 0.7 mmol) and 4-*N*,*N*-diethylamino-2-hydroxybenzaldehyde **26** (0.13 g, 0.7 mmol) in absolute ethanol (7 mL) after refluxing for 48 h to obtain 0.11 g (40%) of orange powder. m.p. 205–208 °C; ^1^H NMR (600 MHz, DMSO-d_6_): δ/ppm = 12.62 (s, 1H, OH), 9.37 (s, 1H, H_arom_), 8.01 (d, 1H, J = 1.09 Hz, H_arom_), 7.74 (d, 1H, J = 8.26 Hz, H_arom_), 7.59 (d, 1H, J = 8.97 Hz, H_arom_), 7.56 (dd, 1H, J_1_ = 1.48 Hz, J_2_ = 8.98 Hz, H_arom_), 6.44 (dd, 1H, J_1_ = 2.29 Hz, J_2_ = 8.99 Hz, H_arom_), 6.18 (d, 1H, J = 2.27 Hz, H_arom_), 4.10 (d, 2H, J = 7.36 Hz, CH_2_), 3.45 (q, 4H, J = 6.97 Hz, CH_2_), 2.20–2.12 (m, 2H, CH), 1.15 (t, 6H, J = 7.08 Hz, CH_3_), 0.90 (d, 6H, J = 6.69 Hz, CH_3_); ^13^C NMR (151 MHz, DMSO-d_6_): δ/ppm = 164.3, 153.9 (2C), 141.6, 138.7, 125.3, 122.6, 120.6, 111.8 (2C), 109.1, 105.9 (2C), 104.1, 97.0 (2C), 50.0, 44.7, 29.4, 20.3, 13.0; Anal. Calcd. for C_23_H_27_N_5_O: C, 70.92; H, 6.99; N, 17.98; O, 4.11. Found: C, 70.97; H, 6.91; N, 17.93; O, 4.08%.

##### (*E*)-1-Isobutyl-2-((4-nitrobenzylidene)amino)-1*H*-benzo[*d*]imidazole-6-carbonitrile **32**

Compound **32** was prepared from 2-amino-6-cyano-1-isobutylbenzimidazole **21** (0.10 g, 0.5 mmol) and 4-nitrobenzaldehyde **27** (0.08 g, 0.5 mmol) in absolute ethanol (3 mL) after refluxing for 24 h to obtain 0.02 g (14%) of yellow powder. m.p. 277–279 °C; ^1^H NMR (300 MHz, DMSO-d_6_) (δ/ppm): 9.66 (s, 1H, H_arom_), 8.41–8.38 (m, 5H, H_arom_), 8.19 (d, 1H, J = 0.98 Hz, H_arom_), 7.88 (d, 1H, J = 8.46 Hz, H_arom_), 7.67 (dd, 1H, J_1_ = 1.47 Hz, J_2_ = 8.38 Hz, H_arom_), 4.31 (d, 2H, J = 7.28 Hz, CH_2_), 2.25–2.13 (m, 1H, CH), 0.90 (d, 6H, J = 6.69 Hz, CH_3_); ^13^C NMR (151 MHz, DMSO-d_6_) (δ/ppm): 165.2, 156.2, 149.8, 140.6, 140.2, 138.2, 130.9 (2C), 125.7, 124.2 (2C), 123.9, 119.8, 112.4, 104.6, 49.6, 29.0, 19.8 (2C); Anal. Calcd. for C_19_H_17_N_5_O_2_: C, 65.69; H, 4.93; N, 20.16; O, 9.21. Found: C, 65.72; H, 4.87; N, 20.09; O, 9.34%.

##### (*E*)-*N*,*N*-Dimethyl-4-(((1-methyl-1*H*-benzo[*d*]imidazol-2-yl)imino)methyl)aniline **33**

Compound **33** was prepared from 2-amino-1-methylbenzimidazole **18** (0.10 g, 0.7 mmol) and 4-*N*,*N*-dimethylamino-benzaldehyde **25** (0.10 g, 0.7 mmol) in absolute ethanol (4 mL) after refluxing for 48 h to obtain 0.01 g (6%) of yellow powder. m.p. 186–188 °C; ^1^H NMR (600 MHz, DMSO-d_6_): δ/ppm = 7.98–7.94 (m, 2H, H_arom_), 7.41–7.36 (m, 2H, H_arom_), 7.32–7.29 (m, 2H, H_arom_), 7.18–7.15 (m, 3H, H_arom_), 3.59 (s, 6H, CH_3_); ^13^C NMR (151 MHz, DMSO-d_6_): δ/ppm = 190.4, 154.7, 132.0, 125.0, 123.8, 123.1, 112.2, 112.1, 111.6, 111.0, 110.4, 19.8, 9.1 (2C); Anal. Calcd. for C_17_H_18_N_4_: C, 71.27; H, 5.65; N, 23.09. Found: C, 71.30; H, 5.61; N, 23.16%.

##### (*E*)-2-((4-(Dimethylamino)benzylidene)amino)-1-methyl-1*H*-benzo[*d*]imidazole-6-carbonitrile **34**

Compound **34** was prepared from 2-amino-6-cyano-1-methylbenzimidazole **22** (0.10 g, 0.6 mmol) and 4-*N*,*N*-dimethylamino-benzaldehyde **25** (0.09 g, 0.6 mmol) in absolute ethanol (3 mL) after refluxing for 48 h to obtain 0.05 g (27%) of yellow powder. m.p. 222–226 °C; ^1^H NMR (400 MHz, DMSO-d_6_) (δ/ppm): 9.27 (s, 1H, H_arom_), 8.01 (d, 1H, J = 1.26 Hz, H_arom_), 7.95 (d, 2H, J = 8.81 Hz, H_arom_), 7.68 (d, 1H, J = 8.32 Hz, H_arom_), 7.58 (dd, 1H, J_1_ = 8.28, J_2_ = 1.37 Hz, H_arom_), 6.84 (d, 2H, J = 8.91 Hz, H_arom_), 3.88 (s, 3H, CH_3_), 3.08 (s, 6H, CH_3_); ^13^C NMR (100 MHz, DMSO-d_6_) (δ/ppm): 166.4, 158.9, 154.3, 141.7, 139.0, 132.8, 125.2, 122.9, 122.8, 122.6, 120.7, 118.2, 112.0, 111.6, 108.8, 104.1, 29.5 (2C), 29.1; Anal. Calcd. for C_18_H_17_N_5_: C, 71.27; H, 5.65; N, 23.09. Found: C, 71.30; H, 5.61; N, 23.16%.

##### (*E*)-5-(Diethylamino)-2-(((1-methyl-1*H*-benzo[*d*]imidazol-2-yl)imino)methyl)phenol **35**

Compound **35** was prepared from 2-amino-1-methylbenzimidazole **18** (0.15 g, 1.0 mmol) and 4-*N*,*N*-diethylamino-2-hydroxybenzaldehyde **26** (0.19 g, 1.0 mmol) in absolute ethanol (7 mL) after refluxing for 24 h to obtain 0.04 g (13%) of orange powder. m.p. 169–173 °C; ^1^H NMR (600 MHz, DMSO-d_6_): δ/ppm = 12.61 (s, 1H, OH), 9.36 (s, 1H, H_arom_), 7.58 (d, 1H, J = 8.87 Hz, H_arom_), 7.54–7.51 (m, 1H, H_arom_), 7.50–7.47 (m, 1H, H_arom_), 7.21–7.15 (m, 2H, H_arom_), 6.42 (dd, 1H, J_1_ = 2.38 Hz, J_2_ = 8.99 Hz, H_arom_), 6.16 (d, 1H, J = 2.26 Hz, H_arom_), 3.78 (s, 3H, CH_3_), 3.45 (q, 4H, J = 7.08 Hz, CH_2_), 1.15 (t, 6H, J = 7.07 Hz, CH_3_); ^13^C NMR (151 MHz, DMSO-d_6_): δ/ppm = 164.5, 163.9, 153.4, 141.9 (2C), 135.8, 122.3, 121.7, 118.3, 110.2, 109.1, 105.5, 97.0, 44.6 (2C), 29.1, 13.0 (2C); Anal. Calcd. for C_19_H_22_N_4_O: C, 70.78; H, 6.88; N, 17.38; O, 4.96. Found: C, 70.71; H, 6.79; N, 17.44; O, 5.03%.

##### (*E*)-2-((4-(Diethylamino)-2-hydroxybenzylidene)amino)-1-methyl-1*H*-benzo[*d*]imidazole-6-carbonitrile **36**

Compound **36** was prepared from 2-amino-6-cyano-1-methylbenzimidazole **22** (0.20 g, 1.2 mmol) and 4-*N*,*N*-diethylamino-2-hydroxybenzaldehyde **26** (0.22 g, 1.2 mmol) in absolute ethanol (10 mL) after refluxing for 48 h to obtain 0.27 g (67%) of yellow powder. m.p. 227–231 °C; ^1^H NMR (400 MHz, DMSO-d_6_) (δ/ppm): 12.40 (s, 1H, OH), 9.37 (s, 1H, H_arom_), 7.99 (d, 1H, J = 1.08 Hz, H_arom_), 7.68 (d, 1H, J = 8.29 Hz, H_arom_), 7.61 (d, 1H, J = 8.97 Hz, H_arom_), 7.57 (dd, 1H, J_1_ = 1.45 Hz, J_2_ = 8.27 Hz, H_arom_), 6.43 (dd, 1H, J_1_ = 2.36 Hz, J_2_ = 9.00 Hz, H_arom_), 6.15 (d, 1H, J = 2.26 Hz, H_arom_), 3.80 (s, 3H, CH_3_), 3.45 (q, 4H, J = 6.98 Hz, CH_2_), 1.15 (t, 6H, J = 6.95 Hz, CH_3_); ^13^C NMR (100 MHz, DMSO-d_6_) (δ/ppm): 165.3, 164.2, 157.6, 153.9, 141.7, 138.9, 125.2, 122.4, 120.7, 111.5, 109.2, 105.9, 104.1, 96.9, 44.7 (2C), 29.5, 13.0 (2C); Anal. Calcd. for C_20_H_21_N_5_O: C, 69.14; H, 6.09; N, 20.16; O, 4.61. Found: C, 69.19; H, 5.98; N, 20.13; O, 4.71%.

##### (*E*)-1-Methyl-2-((4-nitrobenzylidene)amino)-1*H*-benzo[*d*]imidazole-6-carbonitrile **37**

Compound **37** was prepared from 2-amino-6-cyano-1-methylbenzimidazole **22** (0.10 g, 0.5 mmol) and 4-nitrobenzaldehyde **27** (0.08 g, 0.5 mmol) in absolute ethanol (3 mL) after refluxing for 24 h to obtain 0.02 g (14%) of yellow powder. m.p. 277–279 °C; ^1^H NMR (600 MHz, DMSO-d_6_) (δ/ppm): 9.67 (s, 1H, H_arom_), 8.44–8.41 (m, 4H, H_arom_), 8.20 (d, 1H, J = 0.97 Hz, H_arom_), 7.83 (d, 1H, J = 8.38 Hz, H_arom_), 7.69 (dd, 1H, J_1_ = 1.49 Hz, J_2_ = 8.37 Hz, H_arom_), 3.99 (s, 3H, CH_3_); ^13^C NMR (151 MHz, DMSO-d_6_): δ/ppm = 165.9, 156.7, 150.4, 141.2, 140.7, 139.1, 131.5 (2C), 126.2, 124.7 (2C), 124.3, 120.3, 112.6, 105.1, 29.9; Anal. Calcd. for C_16_H_11_N_5_O_2_: C, 62.95; H, 3.63; N, 22.94; O, 10.48. Found: C, 62.91; H, 3.69; N, 22.88; O, 10.41%.

##### (*E*)-*N*,*N*-Dimethyl-4-(((1-phenyl-1*H*-benzo[*d*]imidazol-2-yl)imino)methyl)aniline **38**

Compound **38** was prepared from 2-amino-1-phenylbenzimidazole **19** (0.10 g, 0.5 mmol) and 4-*N*,*N*-dimethylamino-benzaldehyde **25** (0.07 g, 0.5 mmol) in absolute ethanol (3 mL) after refluxing for 48 h to obtain 0.11 g (68%) of yellow powder. m.p. 204–207 °C; ^1^H NMR (400 MHz, DMSO-d_6_) (δ/ppm): 9.29 (s, 1H, H_arom_), 7.76 (d, 2H, J = 8.86 Hz, H_arom_), 1.66–1.57 (m, 5H, H_arom_), 1.54–1.49 (m, 1H, H_arom_), 7.30 (d, 1H, J = 7.89 Hz, H_arom_), 7.28–7.24 (m, 1H, H_arom_), 7.22–7.17 (m, 1H, H_arom_), 6.79 (d, 2H, J = 8.87 Hz, H_arom_), 3.04 (s, 6H, CH_3_); ^13^C NMR (151 MHz, DMSO-d_6_): δ/ppm = 165.5, 156.2, 153.9, 142.1, 135.9, 135.6, 132.3, 129.7 (2C), 128.2, 127.4 (2C), 123.1, 122.9, 122.6, 118.9, 112.0, 110.4; Anal. Calcd. for C_22_H_20_N_4_: C, 77.62; H, 5.92; N, 16.46. Found: C, 77.68; H, 5.98; N, 16.39%.

##### (*E*)-2-((4-(Dimethylamino)benzylidene)amino)-1-phenyl-1*H*-benzo[*d*]imidazole-6-carbonitrile **39**

Compound **39** was prepared from 2-amino-6-cyano-1-phenylbenzimidazole **23** (0.13 g, 0.6 mmol) and 4-*N*,*N*-dimethylamino-benzaldehyde **25** (0.09 g, 0.6 mmol) in absolute ethanol (6 mL) after refluxing for 24 h to obtain 0.04 g (17%) of yellow powder. m.p. 247–251 °C; ^1^H NMR (600 MHz, DMSO-d_6_) (δ/ppm): 9.29 (s, 1H, H_arom_), 8.14 (d, 1H, J = 1.09 Hz, H_arom_), 7.75 (d, 2H, J = 8.97 Hz, H_arom_), 7.66–7.62 (m, 2H, H_arom_), 7.62–7.59 (m, 2H, H_arom_), 7.57 (dd, 1H, J_1_ = 1.47 Hz, J_2_ = 8.28 Hz, H_arom_), 7.56–7.54 (m, 1H, H_arom_), 7.41 (d, 1H, J = 8.27 Hz, H_arom_), 6.78 (d, 2H, J = 8.98 Hz, H_arom_), 3.05 (s, 6H, CH_3_); ^13^C NMR (151 MHz, DMSO-d_6_): δ/ppm = 167.1, 158.6, 154.4, 141.9, 138.6, 135.0, 129.8 (2C), 128.8, 127.5 (2C), 126.1, 123.2, 122.6, 120.4, 112.0, 111.7, 105.0; Anal. Calcd. for C_23_H_19_N_5_: C, 75.59; H, 5.24; N, 19.16. Found: C, 75.65; H, 5.30; N, 19.09%.

##### (*E*)-5-(Diethylamino)-2-(((1-phenyl-1*H*-benzo[*d*]imidazol-2-yl)imino)methyl)phenol **40**

Compound **40** was prepared from 2-amino-1-phenylbenzimidazole **19** (0.10 g, 0.5 mmol) and 4-*N*,*N*-diethylamino-2-hydroxybenzaldehyde **26** (0.09 g, 0.5 mmol) in absolute ethanol (7 mL) after refluxing for 48 h to obtain 0.10 g (52%) of yellow powder. m.p. 197–201 °C; ^1^H NMR (400 MHz, DMSO-d_6_): δ/ppm = 12.40 (s, 1H, OH), 9.37 (s, 1H, H_arom_), 7.67–7.62 (m, 3H, H_arom_), 7.60–7.53 (m, 3H, H_arom_), 7.50 (d, 1H, J = 8.97 Hz, H_arom_), 7.29–7.22 (m, 2H, H_arom_), 7.21–7.17 (m, 1H, H_arom_), 6.39 (dd, 1H, J_1_ = 2.37 Hz, J_2_ = 8.99 Hz, H_arom_), 6.04 (d, 1H, J = 2.27 Hz, H_arom_), 3.41 (q, 4H, J = 7.54 Hz, CH_2_), 1.11 (t, 6H, J = 6.88 Hz, CH_3_); ^13^C NMR (75 MHz, DMSO-d_6_) (δ/ppm): 164.4, 163.5, 153.0, 141.6, 135.6, 135.1, 135.0, 129.7 (2C), 128.2, 126.9 (2C), 122.6, 122.1, 118.3, 109.7, 108.5, 105.0, 96.5, 44.1, 12.5; Anal. Calcd. for C_24_H_24_N_4_O: C, 74.97; H, 6.29; N, 14.57; O, 4.16. Found: C, 74.89; H, 6.19; N, 14.51; O, 4.20%.

##### (*E*)-2-((4-(Diethylamino)-2-hydroxybenzylidene)amino)-1-phenyl-1*H*-benzo[*d*]imidazole-6-carbonitrile **41**

Compound **41** was prepared from 2-amino-6-cyano-1-phenylbenzimidazole **23** (0.20 g, 0.9 mmol) and 4-*N*,*N*-diethylamino-2-hydroxybenzaldehyde **26** (0.16 g, 0.9 mmol) in absolute ethanol (10 mL) after refluxing for 48 h to obtain 0.23 g (66%) of yellow powder. m.p. 111–114 °C; ^1^H NMR (400 MHz, DMSO-d_6_) (δ/ppm): 12.28 (bs, 1H, OH), 9.37 (s, 1H, H_arom_), 8.13–8.11 (m, 1H, H_arom_), 7.68–7.64 (m, 2H, H_arom_), 7.61–7.59 (m, 3H, H_arom_), 7.56 (dd, 1H, J_1_ = 8.28 Hz, J_2_ = 1.56 Hz, H_arom_), 7.51 (d, 1H, J = 9.09 Hz, H_arom_), 7.35 (d, 1H, J = 8.37 Hz, H_arom_), 6.40 (dd, 1H, J_1_ = 9.08 Hz, J_2_ = 2.40 Hz, H_arom_), 6.04 (d, 1H, J = 2.28 Hz, H_arom_), 3.44–3.39 (m, 1H, CH_2_), 1.11 (t, 6H, J = 7.01 Hz, CH_3_); ^13^C NMR (100 MHz, DMSO-d_6_) (δ/ppm): 165.8, 164.4, 156.8, 154.0, 142.0, 138.6, 136.4, 134.7, 130.3, 129.3, 127.8, 126.2, 122.9, 120.4, 111.5, 109.1, 105.9, 105.0, 96.9, 44.7 (2C), 13.0 (2C); Anal. Calcd. for C_25_H_23_N_5_O: C, 73.33; H, 5.66; N, 17.10; O, 3.91. Found: C, 73.36; H, 5.69; N, 17.15; O, 4.02%.

##### (*E*)-4-(((1-Hexyl-1*H*-benzo[*d*]imidazol-2-yl)imino)methyl)-*N*,*N*-dimethylaniline **42**

Compound **42** was prepared from 2-amino-1-hexylbenzimidazole **20** (0.20 g, 0.9 mmol) and 4-*N*,*N*-dimethylamino-benzaldehyde **25** (0.13 g, 0.9 mmol) in absolute ethanol (5 mL) after refluxing for 24 h to obtain 0.09 g (31%) of yellow powder in the form of a mixture of *E*- and *Z*-isomers at a ratio of **42a**/**42b** = 3:1. m.p. 231–234 °C; **42a**: ^1^H NMR (600 MHz, DMSO-d_6_) (δ/ppm): 9.26 (s, 1H, H_arom_), 7.91 (d, 2H, J = 8.88 Hz, H_arom_), 7.55–7.51 (m, 1H, H_arom_), 7.51–7.47 (m, 1H, H_arom_), 7.19–7.14 (m, 2H, H_arom_), 6.84 (d, 2H, J = 8.99 Hz, H_arom_), 4.35 (t, 2H, J = 6.88 Hz, CH_2_), 3.07 (s, 6H, CH_3_), 1.80–1.75 (m, 2H, CH_2_), 1.29–1.17 (m, 6H, CH_2_), 0.79 (t, 3H, J = 7.27 Hz, CH_3_); ^13^C NMR (151 MHz, DMSO-d_6_) (δ/ppm): 165.8, 153.8, 137.8, 124.7, 122.4 (2C), 122.4, 122.2, 120.1, 117.7, 111.5 (2C), 111.1 (2C), 108.4, 103.6, 41.9, 30.5, 28.9, 25.5, 21.9, 13.8; **42b**: ^1^H NMR (600 MHz, DMSO-d_6_) (δ/ppm): 9.67 (s, 1H, H_arom_), 7.69 (d, 2H, J = 8.97 Hz, H_arom_), 7.35 (d, 1H, J = 8.36 Hz, H_arom_), 7.26 (dd, 1H, J_1_ = 1.98 Hz, J_2_ = 6.67 Hz, H_arom_), 7.13–7.08 (m, 2H, H_arom_), 6.79 (d, 2H, J = 8.86 Hz, H_arom_), 4.03 (t, 2H, J = 7.29 Hz, CH_2_), 3.05 (s, 6H, CH_3_), 1.67–1.62 (m, 2H, CH_2_), 1.29–1.17 (m, 6H, CH_2_), 0.84 (t, 3H, J = 6.97 Hz, CH_3_); ^13^C NMR (151 MHz, DMSO-d_6_) (δ/ppm): 156.6, 142.8, 141.2 (2C), 132.1, 123.0, 128.1, 122.4 (2C), 120.7, 117.7 (2C), 111.1, 111.0, 102.0, 41.6, 30.9, 28.2, 25.6, 22.0, 13.8; Anal. Calcd. for C_22_H_28_N_4_: C, 75.82; H, 8.10; N, 16.08. Found: C, 75.78; H, 8.16; N, 16.14%.

##### (*E*)-2-((4-(Dimethylamino)benzylidene)amino)-1-hexyl-1*H*-benzo[*d*]imidazole-6-carbonitrile **43**

Compound **43** was prepared from 2-amino-6-cyano-1-hexylbenzimidazole **24** (0.15 g, 0.6 mmol) and 4-*N*,*N*-dimethylamino-benzaldehyde **25** (0.09 g, 0.6 mmol) in absolute ethanol (5 mL) after refluxing for 48 h to obtain 0.09 g (43%) of yellow powder. m.p. 165–170 °C; ^1^H NMR (300 MHz, DMSO-d_6_) (δ/ppm): 9.27 (s, 1H, H_arom_), 8.02 (d, 1H, J = 0.88 Hz, H_arom_), 7.95 (s, 1H, H_arom_), 7.92 (s, 1H, H_arom_), 7.72 (d, 1H, J = 8.37 Hz, H_arom_), 7.56 (dd, 1H, J_1_ = 1.39 Hz, J_2_ = 8.28 Hz, H_arom_), 7.30 (d, 1H, J = 1.38 Hz, H_arom_), 6.86–6.82 (m, 2H, H_arom_), 4.39 (t, 2H, J = 6.79 Hz, CH_2_), 3.08 (s, 6H, CH_3_), 1.83–1.72 (m, 2H, CH_2_), 1.27–1.20 (m, 6H, CH_2_), 0.77 (t, 3H, J = 6.97 Hz, CH_3_); ^13^C NMR (151 MHz, DMSO-d_6_) (δ/ppm): 165.77, 158.29, 156.63, 153.81, 142.75, 141.22, 137.80, 137.78, 132.15, 129.99, 128.12, 124.68, 122.38, 122.35, 122.20, 120.65, 120.12, 117.68, 111.64, 111.54, 111.14, 111.03, 108.41, 103.57, 101.96, 41.96, 41.60, 30.85, 30.50, 28.90, 28.23, 25.57, 25.50, 21.96, 21.90, 13.80, 13.75; Anal. Calcd. for C_23_H_27_N_5_: C, 73.96; H, 7.29; N, 18.75. Found: C, 74.01; H, 7.24; N, 18.70%.

##### (*E*)-5-(Diethylamino)-2-(((1-hexyl-1*H*-benzo[*d*]imidazol-2-yl)imino)methyl)phenol **44**

Compound **44** was prepared from 2-amino-1-hexylbenzimidazole **20** (0.15 g, 0.7 mmol) and 4-*N*,*N*-diethylamino-2-hydroxybenzaldehyde **26** (0.13 g, 0.7 mmol) in absolute ethanol (5 mL) after refluxing for 48 h to obtain 0.03 g (11%) of yellow powder. m.p. 201–204 °C; ^1^H NMR (600 MHz, DMSO-d_6_) (δ/ppm): 12.68 (s, 1H, OH), 9.36 (s, 1H, H_arom_), 7.56 (d, 1H, J = 8.96 Hz, H_arom_), 7.54–7.48 (m, 2H, H_arom_), 7.21–7.13 (m, 2H, H_arom_), 6.42 (dd, 1H, J_1_ = 2.37 Hz, J_2_ = 8.97 Hz, H_arom_), 6.16 (d, 1H, J = 2.27 Hz, H_arom_), 4.24 (t, 2H, J = 7.07 Hz, CH_3_), 3.44 (q, 4H, J = 7.09 Hz, CH_2_), 1.78–1.73 (m, 2H, CH_2_), 1.29–1.20 (m, 6H, CH_2_), 1.15 (t, 6H, J = 7.08 Hz, CH_3_), 0.80 (t, 3H, J = 7.07 Hz, CH_3_); ^13^C NMR (100 MHz, DMSO-d_6_) (δ/ppm): 164.0, 163.9, 154.6, 154.3, 153.4, 141.9, 135.1, 122.2, 121.8, 118.4, 111.7, 110.2, 109.0, 105.5, 104.9, 97.0, 96.4, 44.6, 44.6, 42.6, 31.2, 29.6, 26.4, 22.5, 14.2, 13.0, 12.9; Anal. Calcd. for C_24_H_32_N_4_O: C, 73.43; H, 8.22; N, 14.27; O, 4.08. Found: C, 73.37; H, 8.09; N, 14.15; O, 4.02%.

##### (*E*)-2-((4-(Diethylamino)-2-hydroxybenzylidene)amino)-1-hexyl-1*H*-benzo[*d*]imidazole-6-carbonitrile **45**

Compound **45** was prepared from 2-amino-6-cyano-1-hexylbenzimidazole **24** (0.15 g, 0.6 mmol) and 4-*N*,*N*-diethylamino-2-hydroxybenzaldehyde **26** (0.11 g, 0.6 mmol) in absolute ethanol (6 mL) after refluxing for 48 h to obtain 0.02 g (8%) of yellow powder. m.p. 233–237 °C; ^1^H NMR (600 MHz, DMSO-d_6_): δ/ppm = 12.47 (s, 1H, OH), 9.38 (s, 1H, H_arom_), 8.00 (d, 1H, J = 1.19 Hz, H_arom_), 7.72 (d, 1H, J = 8.27 Hz, H_arom_), 7.60 (d, 1H, J = 8.99 Hz, H_arom_), 7.56 (dd, 1H, J_1_ = 1.45 Hz, J_2_ = 8.27 Hz, H_arom_), 6.44 (dd, 1H, J_1_ = 2.37 Hz, J_2_ = 8.96 Hz, H_arom_), 6.16 (d, 1H, J = 2.28 Hz, H_arom_), 4.28 (t, 2H, J = 7.06 Hz, CH_2_), 3.45 (q, 4H, J = 6.98 Hz, CH_2_), 1.76–1.72 (m, 2H, CH_2_), 1.28–1.25 (m, 4H, CH_2_), 1.23–1.20 (m, 2H, CH_2_), 1.15 (t, 6H, J = 7.09 Hz, CH_3_), 0.80 (t, 3H, J = 7.08 Hz, CH_3_), ^13^C NMR (151 MHz, DMSO-d_6_): δ/ppm = 164.3, 153.9, 141.7, 138.3, 125.2, 122.6, 120.6, 111.5, 109.1, 105.9 (2C), 104.1, 96.9 (2C), 44.7 (2C), 42.9, 31.1, 29.5, 26.3, 22.4, 14.2, 13.0; Anal. Calcd. for C_25_H_31_N_5_O: C, 71.91; H, 7.48; N, 16.77; O, 3.83. Found: C, 71.94; H, 7.42; N, 16.71; O, 3.88%.

### 4.3. Biology

#### 4.3.1. Antiviral Activity

HEL 299 (ATCC CCL-137; human lung fibroblast), Huh-7 (CLS—300156; human hepatoblastoma), and MDCK (Madin-Darby canine kidney cells; a kind gift from M. Matrosovich, Marburg, Germany) were maintained in Dulbecco’s Modified Eagle Medium (DMEM; Gibco Life Technologies) supplemented with 8% heat-inactivated fetal bovine serum (HyClone, GE Healthcare Life Sciences), 0.075% sodium bicarbonate (Gibco Life Technologies) and 1mM sodium pyruvate (Gibco Life Technologies), and maintained at 37 °C under 5% CO_2_. Antiviral assays towards herpes simplex virus-1 (HSV-1 KOS), human coronavirus (HCoV-229E and -OC43), and respiratory syncytial virus A in HEL 299 cell cultures; sindbis virus, yellow fever virus, Zika virus, and human coronavirus (HCoV-NL63) in Huh-7 cell cultures; and influenza A/H1N1 (A/Ned/378/05), influenza A/H3N2 (A/HK/7/87), and influenza B (B/Ned/537/05) in MDCK cell cultures were performed. On the day of the infection, the growth medium was aspirated and replaced by serial dilutions of the test compounds. The virus was then added to each well and diluted to obtain a viral input of 100 CCID_50_ (CCID_50_ being the virus dose that is able to infect 50% of the cell cultures). Mock-treated cultures receiving solely the test compounds were included in order to determine the cytotoxicity.

After 3 to 7 days of incubation, the virus-induced cytopathogenic effect was measured colorimetrically by the formazan-based MTS cell viability assay (CellTiter 96 AQueous One Solution Cell Proliferation Assay from Promega, Madison, WI), and the antiviral activity was expressed as the 50% effective concentration (EC_50_). In parallel, the 50% cytotoxic concentration (CC_50_) was derived from the mock-infected cells. The activities were compared with the activities of the reference antiviral drugs: remdesivir, ribavirin, zanamivir, rimantadine, and brivudine (BVDU).

#### 4.3.2. Antibacterial Activity

##### Materials

In addition to the synthesized compounds, standard antibiotics ampicillin, ceftazidime, ciprofloxacin, and meropenem from USP were tested. Selected bacterial strains were Gram-negative *E. coli*, *K. pneumoniae*, *A. baumannii* and *P. aeruginosa* and Gram-positive *S. aureus*, *S. pneumoniae,* and *E. faecalis*. The synthesized compounds were prepared as 10 mM DMSO solutions and tested in a final concentration range of 0.2–100 µM [39]. Standard antibiotics were prepared as 5 mg/mL DMSO solutions and tested in a final concentration range of 0.125–64 µg/mL.

##### Methods

Broth microdilution testing was performed according to CLSI (Clinical Laboratory Standards Institute) guidelines. The MIC (minimal inhibitory concentration) value was defined as the last tested concentration of the compound at which there was no visible growth of bacteria. Inoculums for each microorganism were prepared using the direct colony suspension method, where broth solutions that achieved turbidity equivalent to 0.5 McFarland standard were additionally diluted 100× with Ca adjusted MH media (Becton Dickinson, Franklin Lakes, NJ, USA). All of the test plates were incubated for 16–24 h at 37 °C.

MIC values for reference antibiotics against quality control strains were used for confirming the validity of the screen according to the Clinical and Laboratory Standards Institute (CLSI) guidelines. Methods for dilution of antimicrobial susceptibility tests for bacteria that grow aerobically followed M07, 11th edition, 2018, and Clinical and Laboratory Standards Institute (CLSI) guidelines. Performance standards for antimicrobial susceptibility testing followed the M100, 28th edition, 2018.

#### 4.3.3. Cell Culture and Reference Compounds

Human cancer cells used in this manuscript, namely Capan-1, HCT-116, NCI-H460, LN-229, HL-60, K-562, and Z-138, were acquired from the American Type Culture Collection (ATCC, Manassas, VA, USA), while the DND-41 cell line was purchased from the Deutsche Sammlung von Mikroorganismen und Zellkulturen (DSMZ Leibniz-Institut, Braunschweig, Germany). Culture media were purchased from Gibco Life Technologies, Merelbeke, Belgium, and supplemented with 10% fetal bovine serum (HyClone, Cytiva, Marlborough, MA, USA). Vincristine and docetaxel, which were used as the reference inhibitors, were purchased from Selleckchem (Munich, Germany). Stock solutions were prepared in DMSO.

#### 4.3.4. Proliferation Assays

Adherent cell lines LN-229, HCT-116, and NCI-H460 and Capan-1 cells were seeded at a density between 500 and 1500 cells per well, in 384-well tissue culture plates (Greiner, Kremsmünster, Austria). After overnight incubation, the cells were treated with seven different concentrations of the test compounds, ranging from 100 to 0.006 µM.

Suspension cell lines HL-60, K-562, Z-138, and DND-41 were seeded at densities ranging from 2500 to 5500 cells per well in 384-well culture plates containing the test compounds at the same concentration points. The cells were incubated for 72 h with the compounds and were then analyzed using the CellTiter 96^®^ AQueous One Solution Cell Proliferation Assay (MTS) reagent (Promega, Madison, WI, USA) according to the manufacturer’s instructions. The absorbance of the samples was measured at 490 nm using a SpectraMax Plus 384 (Molecular Devices, Silicon Valley, CA, USA), and OD values were used to calculate the 50% inhibitory concentration (IC_50_). The compounds were tested in two independent experiments.

## Data Availability

Not applicable.

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
