# Peer review of "Novel Biologically Active N-Substituted Benzimidazole Derived Schiff Bases: Design, Synthesis, and Biological Evaluation"

_molecules, 2023, doi:10.3390/molecules28093720_

Round 1

Reviewer 1 Report

This paper is focused on the design, synthesis, and biological screening of novel, N-substituted benzimidazole-derived Schiff bases bearing isobutyl, methyl, and n-hexyl side chains or a phenyl group at the N atom on the benzimidazole nuclei as well as a 4-N,N-dimethylamino- or 4-N,N-diethylamino-2-hydroxyphenyl ring attached directly to the imino bond. The paper is well organized, is thorough, and will have a wide readership. 

Regarding formatting in the experimental section, the first letter of each compound described prior to the characterization section should be capitalized: [2-Amino-1-hexylbenzimidazole (21)

Compound 21 was prepared from 10 (1.59 g, 8.3 mmol) and.........].

The authors describe the biological activity across a range of biological targets and compile an SAR; are there any plans to expand the number of analogs in order to develop a QSAR for each general screening assay to optimize lead generation?

In the syntheses, nitro groups are reduced stoichiometrically with SnCl2.  Any reason not to use catalytic hydrogenation? 

References should be uniform with regards to citing page ranges rather than first page of the range.

Author Response

Reviewer 1

This paper is focused on the design, synthesis, and biological screening of novel, N-substituted benzimidazole-derived Schiff bases bearing isobutyl, methyl, and n-hexyl side chains or a phenyl group at the N atom on the benzimidazole nuclei as well as a 4-N,N-dimethylamino- or 4-N,N-diethylamino-2-hydroxyphenyl ring attached directly to the imino bond. The paper is well organized, is thorough, and will have a wide readership. 

1) Regarding formatting in the experimental section, the first letter of each compound described prior to the characterization section should be capitalized: [2-Amino-1-hexylbenzimidazole (21). Compound 21 was prepared from 10 (1.59 g, 8.3 mmol) and.........].

RESPONSE: We have corrected as suggested.

2) The authors describe the biological activity across a range of biological targets and compile an SAR; are there any plans to expand the number of analogues in order to develop a QSAR for each general screening assay to optimize lead generation?

RESPONSE: From this study as well from the earlier published study, we will choose the lead compound/compounds and optimize their structure by using 3D QSAR study (we have many published papers in collaboration with the colleagues who are computational chemists). 3D QSAR study will be used also to predict biological activity to reduce the number of the synthesized compounds. Only the compounds with predicted significant biological activity will be prepared and biologically evaluated.   

3) In the syntheses, nitro groups are reduced stoichiometrically with SnCl2. Any reason not to use catalytic hydrogenation? 

RESPONSE: We have not used the catalytical hydrogenation because we do not have the possibility for this type of reaction in our lab. Furthermore, the reduction with SnCl2 is optimized and used in our research group for many years and products are usually obtained in very good reaction yields.

4) References should be uniform with regards to citing page ranges rather than first page of  the range.

RESPONSE: We have corrected as suggested.

Reviewer 2 Report

Bec and coworkers presented a paper on “Novel biologically active N-substituted benzimidazole derived Schiff bases: design, synthesis, and biological evaluation” but, although it is stated in the title, the described compounds are mostly inactive or very poorly active.

Anyway, it is worthy of note that many diverse biological assays have been carried out.

 Also, the illustrated synthetic procedure is described as a modification of a previous reported protocol, but no references are included in this regard (lines 86-89). What is the novelty or the improvement? You need to clarify this aspect.

I also suggest to accurately revise the Introduction section. It needs to be more concise since, as it is stated, it seems more appropriate for a Review paper rather than for a Research article.

 The authors synthesized many derivatives that need to be characterized more accurately.  For many compounds only the 1H or only the 13C NMR are reported without any Mass data. I also suggest to fully revise the supplementary section; too many typos.

Besides, the manuscript has numerous grammatical and punctuation errors that need to be fixed as a baseline.

For all those reasons, my opinion is that the manuscript is not suitable for Molecules and I don’t recommend the paper for publication.

Author Response

Reviewer 2

Bec and coworkers presented a paper on “Novel biologically active N-substituted benzimidazole derived Schiff bases: design, synthesis, and biological evaluation” but, although it is stated in the title, the described compounds are mostly inactive or very poorly active. Anyway, it is worthy of note that many diverse biological assays have been carried out.

1) Also, the illustrated synthetic procedure is described as a modification of a previous reported protocol, but no references are included in this regard (lines 86-89). What is the novelty or the improvement? You need to clarify this aspect.

RESPONSE: We have added reference 35 which has been already mentioned in the Introduction and added a comment that the compounds were synthesized according to the previously published experimental procedure.

Furthermore, we have indicated in Introduction the novelty of this study:to explore the influence of the side chain attached at the N atom on benzimidazole nuclei, both on antiproliferative and antiviral activity”.

So our attention was to see from the SAR results how the aliphatic chain of different length and types as well as aromatic moiety attached to the nitrogen atom on benzimidazole nuclei will influenced the biological activity. This is also important to conclude if the biological activity will be enhanced or decreased since there won’t be a hydrogen atom attached to the nitrogen atom, which could participate in the additional bonding to biological targets (hydrogen bond for example).Therefore, we have added additional sentence to the Introduction part.     

2) I also suggest to accurately revise the Introduction section. It needs to be more concise since, as it is stated, it seems more appropriate for a Review paper rather than for a Research article.

RESPONSE: We have revised the Introduction part and shortened as much as it was possible.

3) The authors synthesized many derivatives that need to be characterized more accurately. For many compounds only the 1H or only the 13C NMR are reported without any Mass data. I also suggest to fully revise the supplementary section; too many typos.

RESPONSE: We have corrected as suggested. Besides 1H and 13C NMR data, we have also elemental analysis which directly has confirmed the structure of prepared compounds.

4) Besides, the manuscript has numerous grammatical and punctuation errors that need to be fixed as a baseline.

RESPONSE: We have corrected as suggested.

Reviewer 3 Report

Article M. Hranjec et al. is devoted to one of the most common approaches in the search for biologically active compounds involving Schiff bases. There are many works in this area, the synthesis is quite successful in many cases.
Notes. In the introduction, the reference about coordination compounds is too old. It can be refreshed with the following list: 10.1016/j.inoche.2023.110449; 10.1016/j.ica.2022.121323; 10.3390/molecules27227894. The first two works with a biological bias, the latter as an example of obtaining heterometallic compounds using soligands (diketones). Proton spectra of compounds 11, 25, 30 carbon 8, 25 contain a discontinuous baseline. Please record the spectra after recrystallization of the compounds.
In the text you can use and replace in some cases 1H with proton spectra. For some substances, HRMS could be added to confirm the composition of the substances. Figure 2 why is the position of the cyanogroup not indicated? Please redraw the structure for the specific compound.
In general, I recommend a major revision after which a decision can be made to recommend publication.

Author Response

Reviewer 3

Article M. Hranjec et al. is devoted to one of the most common approaches in the search for biologically active compounds involving Schiff bases. There are many works in this area, the synthesis is quite successful in many cases.

1) Notes. In the introduction, the reference about coordination compounds is too old. It can be refreshed with the following list: 10.1016/j.inoche.2023.110449; 10.1016/j.ica.2022.121323; 10.3390/molecules27227894. The first two works with a biological bias, the latter as an example of obtaining heterometallic compounds using soligands (diketones).

RESPONSE: We have added suggested references.

2) Proton spectra of compounds 11, 25, 30 carbon 8, 25 contain a discontinuous baseline.

    Please record the spectra after recrystallization of the compounds.

RESPONSE: We have added corrected NMR spectra and for some compounds the NMR spectra after recrystallization.

3) In the text you can use and replace in some cases 1H with proton spectra.

RESPONSE: We have corrected as suggested.

4) For some substances, HRMS could be added to confirm the composition of the substances.

RESPONSE: We have also elemental analysis which directly has confirmed the structure of prepared compounds, so in our opinion the HRMS is not needed.

5) Figure 2 why is the position of the cyanogroup not indicated? Please redraw the structure for the specific compound.

RESPONSE: Thank you for this observation, it was a mistake. We have corrected and redraw as suggested.

Round 2

Reviewer 2 Report

Bec and coworkers presented an improved version of the paper “Novel biologically active N-substituted benzimidazole derived Schiff bases: design, synthesis, and biological evaluation” but, although they pointed out that the novelty of their study resides in the exploration of the influence of the N-substituents, the manuscript still needs to be revised. In particular, too many grammatical issues are present and, in this regard, an extensive revision for English language is required throughout the entire manuscript.

Moreover, I suggest to include a figure in the Introduction section that illustrates the main molecules cited in the text, in order to improve readability.

Thus, I think that the paper might be reconsidered after a native English speaker carefully proofreads the manuscript. 

Author Response

Bec and coworkers presented an improved version of the paper “Novel biologically active N-substituted benzimidazole derived Schiff bases: design, synthesis, and biological evaluation” but, although they pointed out that the novelty of their study resides in the exploration of the influence of the N-substituents, the manuscript still needs to be revised. In particular, too many grammatical issues are present and, in this regard, an extensive revision for English language is required throughout the entire manuscript.

1) Moreover, I suggest to include a figure in the Introduction section that illustrates the main molecules cited in the text, in order to improve readability.

RESPONSE: We have added 3 examples of previously published Schiff bases in Figure 1.

2) Thus, I think that the paper might be reconsidered after a native English speaker carefully proofreads the manuscript. 

RESPONSE: We have corrected and improved English language by English native speaker as suggested. We do hope that correction will be satisfied for publishing.

Reviewer 3 Report

The corrections made by the authors allow the article to be recommended for publication in the journal Molecules.

Author Response

Thank you for your suggestion.